# Supramolecular Hydrogen Bonding Assembly from Non-Coplanar Aromatic Tetra-^1^H-Pyrazoles with Crystallization-Induced Emission (CIE)

**DOI:** 10.3390/ijms23084206

**Published:** 2022-04-11

**Authors:** Ji Wang, Li-Rong Zhao, Jin Tong, Yan-Min Yu, Xia-Yan Wang, Shu-Yan Yu

**Affiliations:** Center of Excellence for Environmental Safety and Biological Effects, Beijing Key Laboratory for Green Catalysis and Separation, Department of Chemistry and Biology, Beijing University of Technology, Beijing 100124, China; jw@126.com (J.W.); lrz@163.com (L.-R.Z.); ymy@bjut.edu.cn (Y.-M.Y.); xyw@bjut.edu.cn (X.-Y.W.); syy@bjut.edu.cn (S.-Y.Y.)

**Keywords:** supramolecular assembly, hydrogen bonding, aromatic multi-^1^H-pyrazoles, crystallization-induced emission (CIE)

## Abstract

Here, we report a design strategy for constructing supramolecular organic frameworks by introducing ^1^H-pyrazole groups to aromatic cores as non-coplanar molecules to form diverse supramolecular assemblies through multiple ^1^H-pyrazole [N−H···N] hydrogen bonds as well as other weak interactions. The new supramolecular organic frameworks displayed interesting crystallization-induced emission (CIE) behavior.

## 1. Introduction

Making use of flexibility and reversibility of hydrogen bonding interactions and upon meticulous design of building blocks with abilities in the form of supramolecules, controllably supramolecular organic hydrogen assemblies with satisfying architectures and interesting properties could be designed and constructed [1,2,3]. Therefore, supramolecular complexes self-assembled through intermolecular hydrogen-bonding interactions, named HOFs, have been proven to be a potentially tunable platform for constructing functional materials [4,5,6,7]. So far, various supramolecular HOFs, made by employing organic molecules, have been reported and employed in fields like sensing, capture, separation, proton conductivity, catalysis, and so on [8,9,10,11,12,13,14].

However, establishment of a universal design strategy and creation of new functionalities are still in demand. Structural determination helps to gain deep insight into the structure−property relationships at the molecular level. The structures and properties of HOFs can be tuned by changing the aromatic linkers and hydrogen bonds units. The hydrogen bonds units such as −COOH, pyrazole, −CN, pyridine, imidazole etc., and aromatic linkers such as tetraphenylethylene, benzene, pyrene, anthracene, triphenylamine, etc., can control the configuration of HOFs, and then further display a great influence on properties [15,16]. Recently, the synthesis of stable HOFs with well-defined structure and luminescence has drawn intense attention of researchers [9,17,18]. Intermolecular interactions, such as hydrogen-bonding and π···π stacking, have a close relation to the luminescence properties of π-conjugated molecules as building blocks to construct supramolecular organic complexes [19,20,21].

Although the aromatic-rich linkers featuring multi-^1^H-pyrazole motifs as fundamental building blocks are promising in the construction of supramolecular assemblies as hydrogen-bonded organic frameworks via hydrogen-bonding [N–H···N] interaction [22,23,24], such HOFs based on the non-coplanar tetra-^1^H-pyrazoles remain unexplored despite their interesting structures and optical properties [25]. We have prepared three tetra-^1^H-pyrazolyl-substituted compounds named as **1**–**3** [H_4_TP-Be (**1**) H_4_TP-Py (**2**) H_4_TP-TPE (**3**), TP = tetrapyrazole Be = benzene Py = pyrene TPE = tetraphenylethylene] bearing four ^1^H-pyrazole binding sites as flexible arms (Figure 1a and Appendix A) with dramatically different supramolecular self-assembly abilities from ^1^H-pyrazole units and photoluminescence characteristics from benzene/pyrene/tetraphenylethylene cores [26].

In this work, we decided to explore the hydrogen bonding and π···π stacking induced self-assembly of non-planar aromatic molecules with interesting optical properties that contain aromatic cores and multi-^1^H-pyrazole moieties, since ^1^H-pyrazole is a poor metal-binding ligand that still participates very strongly in hydrogen bonding due to the adjacent amine-and imine-like N atoms therein. And the aromatic cores benzene, pyrene, and tetraphenylethylene were chose for comparison with each other in the solid structures and photoluminescence properties.

## 2. Results and Discussion

We employed these tetra-^1^H-pyrazolyl-substituted compounds **1**–**3** (Figure 1a) to develop supramolecular organic porous frameworks **HOF-1**, **HOF-2**, and **HOF-3** via [N−H···N] hydrogen bonds between the terminal ^1^H-pyrazole rings, π···π stacking between the aromatic rings, and C−H···π weak intermolecular interactions. Different aromatic cores give various crystal packing models in the crystal state which control the photofluorescence of these crystal materials from aggregation-induced emission (AIE) to crystallization-induced emission (CIE). Finally, a promising generation of optical crystal materials was developed by controlling the production of supramolecular organic porous crystals via self-assembly of non-coplanar aromatic multi-^1^H-pyrazoles as CIE luminogens.

The single crystals of supramolecular structures as HOFs (**HOF-1**, **HOF-2**, and **HOF-3**) are obtained by diffusion of isopropyl ether into compound’s solutions in DMA for several weeks, and characterized by X-ray crystallography (Figure 1b). The results show that the three new aromatic linkers are not flat; the ^1^H-pyrazole moieties and the aromatic cores are twisted. As shown in Figure 1, the dihedral angles between the ^1^H-pyrazole groups and the aromatic core are 39.68° (A_1_-B, A_3_-B) and 46.59° (A_2_-B, A_4_-B) for **1**, 35.37° (A_1_-B_1_, A_3_-B_3_) and 40.77° (A_2_-B_2_, A_4_-B_4_) for **2**, 3.41° (A_1_-B_1_, A_4_-B_4_) and 5.93° (A_2_-B_2_, A_3_-B_3_) for **3**, respectively. And the dihedral angles between benzene rings (B_1_, B_2_, B_3_, B_4_) of **3** range from 44.73 to 87.59°.

Single-crystal X-ray diffraction indicates that aromatic tetra-^1^H-pyrazole linkers in the structures **HOF-1** and **HOF-2** connect with four neighboring linkers through four single hydrogen bond N−H···N from ^1^H-pyrazole units to form regular tetragonal honeycomb hydrogen-bonded frameworks, which is the result of the symmetry of the ligands (Figure 1a, Figure 2, and Figure 3a). The N···N distance and the N−H···N angle are 2.855 Å and 158.41° of **HOF-1**, and 2.890 Å and 157.83° of **HOF-2**, respectively, falling into strong hydrogen bond range, according to literature at 2.49 to 3.15 Å [26]. Each 2D square layer interacts with adjacent layers through π···π stacking interactions as shown in Figure 1d and Figure 3f and with an interlayer distance of 3.713 Å for **HOF-1** and 3.708 Å for **HOF-2** (Appendix A), stacking as AA mode to form a one dimensional square channels of 13.267 × 8.179 Å for **HOF-1** and 14.457 × 14.622 Å for **HOF-2** (Figure 1e,f and Figure 3e). As shown in Figure 2, the surface of the channel is very smooth in **HOF-1**. Notably, the channel surface of **HOF-2** is relatively undulated and the layers of **HOF-2** are themselves simultaneously interwoven to form extremely complicated two-dimensional (2D) layer structures with the result that the void of the channel is occupied by H_4_TP-Py molecules through π···π stacking interactions as shown in Figure 3f. Entanglement by interpenetration and polycatenation among square layers stabilized the structures due to increasing molecular packing density and additional intermolecular interactions.

However, in the solid-state structure of **HOF-3**, the H_4_TP-TPE molecule has a distorted tetrahedral configuration, which can be attributed to rotation of the C=C bond among the four phenyl rings of the TPE moiety. H_4_TP-TPE exhibited a typical propeller-like structure in the solid state. This propeller-like structure is beneficial to the RIR process in the aggregated state and could effectively weaken the π···π stacking interaction to avoid fluorescence quenching. Unlike in the structures of **HOF-1** and **HOF-2**, the cross-shaped dimer unit in the structure of **HOF-3** is observed via C−H···π interacted between molecules as seen in Figure 4a. Additionally, complicated 2D frameworks are formed in the stacking structure (Figure 4b). Furthermore, the bulk crystallinity of **HOF-1**, **HOF-2**, and **HOF-3** were confirmed by powder X-ray diffraction analysis (Appendix A).

On the basis of the observations of photophysical properties of compounds **1**–**3** in the dilute solution and aggregation state (Appendix A), we further investigated their optical properties in the crystalline state as depicted in Figure 5. The photoluminescent (PL) spectroscopic measurements revealed that the emission of **HOF-1** is slightly enhanced at 375 nm under excitation at 306 nm upon crystallization (photoluminescent quantum yield (PLQY) = 0.059 in DMA solution; 0.063 in the microcrystal state). Although the PL spectra revealed that compound **2** showed a strong emission in dilute DMA under excitation at 450 nm (PLQY = 0.734), the microcrystals as **HOF-2** showed a weak and broad emission centered at 525 nm (PLQY = 0.049). Interestingly, the emission efficiency of **HOF-3** displayed a sharply increase compared to the free compound **3** in dilute solution with PLQY change from 0.330 to 0.030, which can be regarded as typical crystallization-induced emission (CIE) behavior. The distinct difference in crystalline state PLQY suggested a significant effect of molecular packing. Thus, the decrease in emission efficiency from the **HOF-2** results from the strong π···π stacking. In contrast, the propeller-like structure without π···π stacking interactions in **HOF-3** suggested that its CIE character originated from the RIR process.

## 3. Materials and Methods

The syntheses and characterizations of compounds **1**, **2**, and **3** were found in SI according to the literature [26]. Dimethylacetamide and diisopropyl ether were freshly distilled before use. The hand-held UV lamp (Spectroline ENF260C/FC; wavelength: 320–400 nm, peak at 365 nm) was used as the photoexcitation source—UV/Vis spectrophotometer. Fluorescence spectra were recorded on a Hitachi-F7000 fluorescence spectrophotometer. Absolute luminescence quantum yields were measured by Hamamatsu absolute PL quantum yield spectrometer FLS-1000. Powder X-ray diffraction patterns (PXRD) were recorded on a BRUKER D8-Focus Bragg-Brentano X-ray powder diffractometer equipped with a Cu sealed tube (λ = 1.54178 Å) at room temperature. X-Ray diffraction data of the crystals of compounds **HOF-1**–**HOF-3** were collected at 293 K on a Bruker Apex CCD and Bruker D8 QUEST area detector equipped with a graphite monochromated MoKα radiation (λ = 0.71073 Å). The structure was solved by direct methods and refined employing full-matrix least-squares on F2 by using SHELXTL (Sheldrick, 2014, Bruker, Germany) program and expanded using Fourier techniques [27,28]. All non-H atoms of the complexes were refined with anisotropic thermal parameters. The hydrogen atoms were included in idealized positions. The molecules of **HOF-1–HOF-3** form an H-bond framework. The final residuals along with unit cell, space group, data collection, and refinement parameters are presented in Appendix A.

## 4. Conclusions

In summary, we synthesized and characterized three novel multi-^1^H-pyrazole-based novel supramolecular hydrogen organic frameworks by self-assembly during crystallization employing ^1^H-pyrazole-based [N−H···N] hydrogen bonding, C−H···π, and π···π stacking weak interactions. The new supramolecular hydrogen framework based on tetraphenylethene building blocks exhibit excellent crystallization-induced emission (CIE) phenomenon. This prospect provides a strategy to the controllable synthesis of novel supramolecular hydrogen organic networks and highlights the important role of aromatic cores in the ligands with optical properties.

## Data Availability

The following supporting information can be downloaded at: https://www.mdpi.com/ethics.

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
