# Peer review of "Supramolecular Hydrogen Bonding Assembly from Non-Coplanar Aromatic Tetra-1H-Pyrazoles with Crystallization-Induced Emission (CIE)"

_ijms, 2022, doi:10.3390/ijms23084206_

Round 1

Reviewer 1 Report

This paper authored by Ji Wang and coworkers titled “Supramolecular Hydrogen Bonding Assembly from Non-coplanar Aromatic Tetra-1H-pyrazoles with crystallization-induced 3 emission (CIE)” describe a study of the properties of new compounds that contain aromatic cores and multi-1H-pyrazole moieties. The paper is written reasonably well and good structured, the bibliography cited in the introduction is adequate. the results are relevant and interesting and overall, they are supported by the experimental data obtained. Although the synthesis of the compound is indicated by the reference used, I suggest the authors show the synthesis and characterization of the compounds synthesized and revise Supporting Information, contents point number 5 is not appears, this must be improved. I recommend to accept the manuscript after minor revision.

Author Response

Thank you very much for you careful work and nice reading. W have done the corrections according to your comments.

Reviewer 2 Report

The manuscript submitted by Yu et al. presented the assembly features of three aromatic tetra-1H-pyrazoles compounds. Crystals of all compounds were obtained and studied. Among the three, HOF-3 showed excellent crystallization-induced emission (CIE) behavior and could be potential functional materials. Although the research is interesting, there are still some issues that need to be addressed.

  1. The figure numbers are messy, making reading difficult:

1) Line 79, ‘As shown in Fig. S1, the dihedral angles between…’, there are only optimized computational structures in Fig. S1 and the values could not match the ones presented in the manuscript.

2) Line 88-89, ‘which is the result of the symmetry of the ligands (Fig. 1a, 2 and 3a)’, where is Fig. 2?

3) Line 92, ‘through π···π stacking interactions as shown in Fig. d and 2f’, what is Fig. d and where is Fig. 2f?

4) Line 95-102, again, where is Fig. 2 and I also could not find Fig. 3h. I could not understand what this paragraph is talking about without finding these two figures.  

  1. Line 83-84, what is the point to include the computational results since crystal structures for all three compounds have been obtained? If the author wants to compare the crystal structure and the computational structure, is there any explanation to ‘The observed distortion is larger than that of the optimized structure’?
  2. In the crystal structure of HOF-3, is there any inter-molecular hydrogen-bonding interaction between the cross-shaped dimer unit?
  3. The author should explain how the bulk crystallinity of HOF-1, HOF-2 and HOF-3 were confirmed by powder X-ray diffraction analysis. In Fig S4 - S6, the experimental results don’t look similar to the simulated results.

Author Response

Thank your four your careful wrok and nice suggestions. We have done the corrections according to you comments.

Round 2

Reviewer 2 Report

All the comments have been addressed and I recommend accepting the manuscript after carefully checking the text editing. (eg. line 144, 'syngheses' should be 'syntheses')